# Chronic Low Back Pain: A Narrative Review of Recent International Guidelines for Diagnosis and Conservative Treatment

**DOI:** 10.3390/jcm12041685

**Published:** 2023-02-20

**Authors:** Vanina Nicol, Claire Verdaguer, Camille Daste, Hélène Bisseriex, Éric Lapeyre, Marie-Martine Lefèvre-Colau, François Rannou, Alexandra Rören, Julia Facione, Christelle Nguyen

**Affiliations:** 1Service de Rééducation et de Réadaptation de l’Appareil Locomoteur et des Pathologies du Rachis, Hôpital Cochin, AP-HP, Centre-Université de Paris, 75014 Paris, France; 2Service de Médecine Physique et de Réadaptation, Hôpital d’Instruction des Armées Percy, 92140 Clamart, France; 3UFR de Médecine, Faculté de Santé, Université Paris Cité, 75006 Paris, France; 4INSERM UMR-S 1153, Centre de Recherche Épidémiologie et Statistique Sorbonne Paris Cité, 75004 Paris, France; 5Service de Médecine Physique et de Réadaptation, Hôpital d’Instruction des Armées Clermont-Tonnerre, 29240 Brest, France; 6Defence Rehabilitation Post-Traumatic Center, Hôpital d’Instruction des Armées Percy, 92140 Clamart, France; 7INSERM UMR-S 1124, Toxicité Environnementale, Cibles Thérapeutiques, Signalisation Cellulaire et Biomarqueurs (T3S), Campus Saint-Germain-des-Prés, 75006 Paris, France; 8Département Universitaire des Sciences de la Rééducation et de la Réadaptation, Faculté de Santé, Université Paris Cité, 75006 Paris, France

**Keywords:** chronic low back pain, guidelines, diagnosis, treatments

## Abstract

Chronic low back pain (cLBP) is a public and occupational health problem that is a major professional, economic and social burden. We aimed to provide a critical overview of current international recommendations regarding the management of non-specific cLBP. We conducted a narrative review of international guidelines for the diagnosis and conservative treatment of people with non-specific cLBP. Our literature search yielded five reviews of guidelines published between 2018 and 2021. In these five reviews, we identified eight international guidelines that fulfilled our selection criteria. We added the 2021 French guidelines into our analysis. Regarding diagnosis, most international guidelines recommend searching for so-called yellow, blue and black flags, in order to stratify the risk of chronicity and/or persistent disability. The relevance of clinical examination and imaging are under debate. Regarding management, most international guidelines recommend non-pharmacological treatments, including exercise therapy, physical activity, physiotherapy and education; however, multidisciplinary rehabilitation, in selected cases, is the core treatment recommended for people with non-specific cLBP. Oral, topical or injected pharmacological treatments are under debate, and may be offered to selected and well-phenotyped patients. The diagnosis of people with cLBP may lack precision. All guidelines recommend multimodal management. In clinical practice, the management of individuals with non-specific cLBP should combine non-pharmacological and pharmacological treatments. Future research should focus on improving tailorization.

## 1. Introduction

Low back pain (LBP) is a public and occupational health problem that is a major professional, economic and social burden. Up to 84% of the general population will experience an episode of LBP during its life time, and recurrence rates are high [1]. Acute LBP is the second reason for consultations in general medicine, and chronic LBP is the eighth [ref]. One in five LBP episodes result in sick leave. LBP represents 30% of sick leaves that longer than 6 months, and 20% of work accidents. LBP has become the leading cause of exclusion from work before the age of 45, and the third cause of work disabilities in France [2].

Non-specific LBP is defined as axial/non-radiating pain occurring primarily in the back, with no signs of a serious underlying condition (such as cancer, infection or cauda equina syndrome), spinal stenosis, radiculopathy or another specific spinal cause (such as vertebral compression fracture or ankylosing spondylitis) [3,4]. The diagnosis of non-specific LBP implies no known pathoanatomical cause [3]; however, LBP is a symptom, not a diagnosis. Without defining a precise pathoanatomical cause, there is little rationale for intervention [5,6]. This may, in part, explain difficulties to manage “non-specific LBP” and the persistent burden of chronic LBP [6]. The international LBP guidelines highlight a different approach that relies on more precise phenotyping of biopsychosocial factors, in order to provide a more effective treatment, prevent chronicization and address the burden of LBP in a more rationale manner.

In the present narrative review of international guidelines for the diagnosis and conservative treatment of people with non-specific LBP, we aimed to provide a critical overview of current international recommendations regarding the management of non-specific LBP, focusing on its chronic stage, and based on high-quality evidence.

## 2. Materials and Methods

We searched the PUBMED database on June 2021 for reviews of guidelines using the following key words: “low back pain” AND “guidelines” AND “review” OR “overview”. We selected guidelines that were included in these reviews when information was reported regarding either the diagnosis or the treatment of non-specific LBP, focusing on the chronic phase, available in English or French, published after 2015, and considered to be of moderate to high quality according the AGREE II tool. We did not select guidelines from the acupuncture, osteopathic or chiropractic associations.

## 3. Results

### 3.1. Literature Search

Overall, our PUBMED search yielded five reviews of guidelines published between 2018 and 2021 [7,8,9,10,11] (Figure 1). In these five reviews, we identified seven international guidelines that fulfilled our selection criteria [4,12,13,14,15,16,17,18]. We added the 2021 French guidelines, since they were available in the PUBMED database and were based on high-quality evidence [2].

### 3.2. Diagnosis (Table 1)

*Relevance of clinical examination.* According to symptom duration, LBP can be defined as acute (less than 2 to 4 weeks), subacute (from 4 to 12 weeks) or chronic (more than 12 weeks). In the 2021 French guidelines, the term acute LBP flare-up, rather than acute LBP, was suggested to reflect the recurrence of symptoms in the patient’s symptoms trajectory, with or without a background of chronic LBP, because acute LBP flare-ups may require temporary intensification of treatments [2]. The Canadian, French and German guidelines introduced the notion of recurrence of LBP [2,13,14], defined in the 2021 French guidelines as the recurrence of LBP within 12 months, which is also a risk for chronicity [2]. Almost all of the guidelines underlined the importance of assessing early psychosocial factors (yellow flags) from the initial phase (after about 2 or 4 weeks), in order to stratify the risk for chronicity, and to establish risk-based management. Composite questionnaires, such as the STarT Back screening tool (stratified management) [19] or the Örebro Musculoskeletal Pain Screening Questionnaire (absenteeism prognosis) [20], can be used to assess the risk of chronicity. The 2021 French guidelines also suggested assessing others contributors to chronicity, including fears and beliefs and psychological and social distress (black flags, blue flags), as well as beliefs related to physical activity and work with LBP [21] or the Hospital Anxiety and Depression scale [22].
jcm-12-01685-t001_Table 1Table 1Diagnosis.AuthorYearCountryRed FlagsPsychosocial Risk FactorHistory and Physical ExaminationImagingvan Wambeke [18]2017BelgiumAssess for signs of serious underlying conditions including cancer, infection, trauma, inflammatory or severe neurological impairments (e.g., cauda equina syndrome)Search for differential diagnoses particularly for new or changed symptomsConsider using screening tools for risk stratification (e.g., STarT Back or Örebro) for new episodes from 48 h after the pain onset. Risk stratification is aimed at informing shared decision making about stratified managementThere is insufficient evidence to recommend for or against specific clinical tests, because no test considered in isolation has adequate sensitivity and specificity for determining the cause of painThe objective of history taking and physical examination is to assess for signs of serious underlying conditionImaging should not be routinely offered in the absence of red flagsConsider prescribing imaging if expected results may lead to change managementExplain to patients with low back pain that imaging may not be necessaryTOP [13]2015CanadaAssess for signs of serious underlying conditions requiring specific evaluation and treatmentSearch for surgical emergency (e.g., cauda equina syndrome)Assess for psychosocial risk factors (i.e., yellow flags including include fear, financial problems, anger, depression, job dissatisfaction, family problems or stress)Conduct a review of these factors if there is no improvementThere is insufficient evidence to recommend for or against using screening tools for risk stratificationThere is insufficient evidence to recommend for or against using the Clinically Organized Relevant Exam (CORE)Lumbar spine X-rays are poor indicators of serious underlying conditions. In the absence of red flags, spinal and lumbar spine X-rays are not recommendedSpecific and appropriate diagnostic imaging should be selected on the basis of the condition being soughtLumbar spine X-rays may be considered prior to other diagnostic imaging to assess stability and stenosis (e.g., MRI): views should be limited to standing antero-posterior and lateral views. MRI scanning has limited value in the absence of red flags, radiculopathy or neurogenic claudicationCT scans may be considered when vertebral fractures are suspected, or MRI contraindicatedChenot [14]2017GermanyAssess for signs requiring specific imaging or laboratory tests and/or referral to a specialistAssess for psychosocial and workplace risk factors from the beginningConsider using screening tools for psychosocial (i.e., yellow flags) and workplace risk factors from 4 weeks after the pain onset if pain persists despite adequate treatment (i.e., provided in accordance with guidelines)The objective of history taking and physical examination is to assess for signs of a dangerous course of the disease or serious underlying conditionWhen such signs are absent, no further diagnostic steps should be undertaken, because they will exceptionally result in a specific diagnosis, and may promote chronic painCurrent evidence does not support routine imagingIndication for diagnostic imaging should be reassessed from 4 to 6 weeks after the pain onset if pain or activity limitations persist despite adequate treatment (i.e., provided in accordance with guidelines)Indication for diagnostic imaging may be reassessed earlier, from 2 to 4 weeks after the pain onset, if a currently employed patient has been unable to work for a long period of time, or if a diagnostic evaluation is required before multimodal treatmentImaging that lacks any potential therapeutic relevance should be avoidedHAS [2]2019FranceAssess for signs of underlying conditions requiring specific and/or urgent care in case of recent lumbar pain or worsening of symptoms or new symptoms (i.e., acute flare-up of low back pain or change in symptoms)Assess early for psychosocial risk factors (i.e., yellow flags). Fears and beliefs, psychological and social contexts must be identified earlyConsider using screening tools for risk stratification (e.g., STarT Back or Örebro) to assess the risk for chronic pain. Other specific questionnaires assessing the level of fears and avoidances (e.g., FABQ) or symptoms of anxiety and depression (e.g., HADS) can also be usedAssess for risk factors of prolonged inability to work and/or to return to work (i.e., blue flags and black flags) in the event of repeated or prolonged (>4 weeks) sick leave. Consider requesting the expertise of an occupational physician in this caseNo dataIt is recommended to explain to the patient why imaging is not necessary in the first place, and if there is absence of systematic correlation between the symptoms and the radiological signsIn the absence of a red flag, spinal imaging (i.e., MRI or a CT scan if MRI is contra-indicated) should be considered if pain persists beyond 3 months, or if an invasive procedure (epidural infiltration or spinal surgery) is plannedIn the absence of a red flag, there is no indication to perform isolated X-rays, except to asses for instability or spinal deformityThere is no indication to repeat imaging in the absence of changes in symptomsNICE [12]2016UKAssess for alternative conditions, particularly for new or changed symptomsSearch for specific causes of low back pain including cancer, infection, trauma or inflammatory diseaseConsider using screening tools for risk stratification (e.g., STarT Back), at first point of contact with a healthcare professional, for each new episode of low back pain, in order to inform shared decision making about stratified managementNo dataImaging should not be routinely offeredExplain to patients that they may not need imagingConsider imaging in specialist settings of care, only if the result is likely to change managementQaseem [17]2017USANo dataNo dataNo dataNo dataNASS [16]2020North AmericaNo dataAssess for psychosocial and workplace risk factors for chronic painConsider previous episodes of low back pain as a prognostic factor for chronic pain Consider pain severity and functional impairment to stratify the risk for chronic painConsider psychosocial factors as prognostic factors for return to work following an episode of acute low back pain Consider a nonstructural cause of low back pain in patients with diffuse low back pain and tendernessUsing fear avoidance behavior to determine the likelihood of a structural cause of low back pain There is insufficient evidence to recommend for or against using diffuse low back tenderness to predict the presence of disc degenerationThere is insufficient evidence to recommend for or against an association between low back pain and spondylosisThere is insufficient evidence to recommend for or against imaging in the absence of a red flag to recommend for or against imaging findings correlating with low back painThere is insufficient evidence to determine whether imaging findings contribute to decision making to guide treatmentACOEM [23]2020USAAssess for red flags through medical history and physical examinationAssess for psychosocial risk factors at follow-up visitsPhysical examination includes straight leg raising test and neurological examination.Assess for nerve root compression by MRI or CT-scan in patients with symptoms that are not improving over 4 to 6 weeks with signs of nerve root dysfunctionVA/DOD [4]2017USAAssess for neurologic deficits through medical history and physical examination (e.g., radiculopathy, neurogenic claudication)Assess for signs of serious underlying conditions including malignancy, fracture, infectionPerform mental health screening to inform selection of treatmentHistory taking and physical examination are critical to identify treatable causes of low back painDiagnostic imaging may be considered in patients with serious or progressive neurologic deficits, or when a red flag is presentThere is insufficient evidence to recommend for or against imaging in patients with pain for longer than 1 month who have not improved or responded to initial treatments

The diagnostic approach for acute LBP is well codified. International guidelines agree on the interest of identifying warning signs (red flags) with any acute LBP flare-up, symptom aggravation or new symptom appearance, pointing to an underlying pathology requiring specific and/or urgent management (i.e., traumatic or tumor cause, infectious or inflammatory disease). The diagnostic approach for chronic LBP is less consistent. In the literature, a diagnosis of non-specific chronic LBP implies no known serious pathoanatomical cause. Some international guidelines highlighted the need to more precisely phenotype chronic LBP, in order to better understand origins of symptoms, and to offer more targeted and effective treatments [2,16,23]. The 2019 US Veteran Affairs guidelines suggested less clearly that LBP could be related to sacroiliac joint disorders and spinal stenosis [4]. The 2019 American College of Occupational and Environmental Medicine (ACOEM) guidelines were sought for the treatment of various spinal disorders including LBP, sciatica/radiculopathy, spondylolisthesis, facet osteoarthritis, degenerative disc disease, failed back surgery syndrome and spinal stenosis [23]. Consistently, the 2021 French guidelines distinguished “non-degenerative LBP” (formerly known as “specific LBP”), “degenerative LBP” supposedly related to discogenic, facet, ligamentous, muscular or mixed causes, regional or global spinal malalignment, and “LBP unrelated to anatomical lesions” [2]. Finally, there is inconclusive evidence to recommend for or against using their Clinically Organized Relevant Exam back tool for chronic low back pain [13]. Despite phenotyping LBP being usually considered useful to advance the diagnosis of LBP, no specific recommendations have been made in international guidelines regarding clinical examination, including medical history or physical tests.

*Relevance of imaging*. International guidelines agree that in the presence of red flags, or if an invasive procedure (e.g., epidural injection or spinal surgery) is considered, spinal imaging (MRI or CT-scan if MRI is contraindicated) is recommended. In the absence of red flags, there is no indication to perform spinal imaging in the case of LBP acute flare-ups, recurring LBP or no new symptoms appearing. In case of chronic LBP, the relevance of imaging is under debate. Most guidelines do not recommend spinal imaging, because correlations between symptoms and radiological signs are often lacking, and may promote unnecessary treatment and chronicization. Only the 2021 French guidelines recommend MRI for chronic LBP longer than 3 months [2]. In the absence of red flags, international guidelines agree that X-rays have limited interest in the diagnosis of LBP. Only the Canadian and French guidelines recommend X-rays to evaluate spinal instability (i.e., spondylolisthesis) and/or spinal alignment [2,13].

### 3.3. Treatments (Table 2)

#### 3.3.1. Non-Pharmacological Treatments

*Exercise therapy and physical activity*. Although there is insufficient evidence that outcomes from a home-based exercise program are different than no care [16], all international guidelines recommend physical exercise. The French and German guidelines recommend maintaining usual physical activities [2,14]. Concerning the modalities of the physical exercises that practitioners have to recommend to their patients, there is no consensus. A combination of approaches seems to be relevant [18], as well as taking into account people’s specific needs, preferences and capabilities when choosing the type of exercise [12]. The Canadian guidelines favor gentle exercise, and a gradual increase in the exercise level within pain tolerance, specifying that when exercise exacerbates pain, the program should be assessed by a qualified physical therapist, and if exercise still exacerbates pain, patients should be assessed by a physician [13]. No type of activity seems to be superior to another, but certain types of activities are more regularly cited in the recommendations. For example, aerobic exercise is repeatedly and strongly recommended by the ACOEM [15]. The North American Spine Society (NASS) recalls that aerobic exercises improve pain, disability and mental health in patients with non-specific LBP at short-term follow-up, even if there is insufficient evidence of an improvement at the long-term follow-up [16]. Water-based exercise therapy could be offered for selected chronic LBP patients (e.g., extreme obesity, significant degenerative joint disease) [15]. Exercises including Pilates, yoga and Tai Chi are frequently recommended. Yoga may offer medium-term improvements in pain and function compared to usual care [16], but for selected and motivated patients [15]. Stretching is controversial in the absence of a significantly reduced range of motion [15]. There is no consensus to favor individual or group sessions.
jcm-12-01685-t002_Table 2Table 2Treatments.AuthorYearCountryPharmacological TreatmentNon Pharmacological Treatment


General TreatmentTopical TreatmentSpinal InjectionPhysical ActivityPhysiotherapyInformation/EducationPsychotherapyMultidisciplinary TreatmentOther Treatmentsvan Wambeke [18]2017Belgium**Recommended, if a medication is required**: - first line: oral NSAIDs - second line: weak opioids ± acetaminophen **Not recommended**: - acetaminophen as a single medication nor opioids in routine - selective serotonin-norepinephrine reuptake inhibitors - tricyclic antidepressants or non-selective serotonin-norepinephrine reuptake inhibitors in routine - anticonvulsants- antibiotics, muscle relaxants **No clear recommandation**: topical NSAIDs**Not recommended**: non epidural spinal injections**No clear recommandation**: facet joint infiltration **Recommended**: exercise programme (specific exercises or a combination of approaches) **Recommended**: manipulation, mobilization or soft tissue techniques: only as part of a multimodal treatment with a supervised exercise program**Recommended**: - provide advice and information to help self-management - promote and facilitate return to work or normal activities of daily living as soon as possible**Recommended**: psychological intervention using a cognitive behavioral approach: - only as part of a multimodal treatment with a supervised exercise program- optional and depending on patients risk stratification**Recommended**: multidisciplinary rehabilitation program which combines physical and psychological component (cognitive behavioral approach, takes into account the person’s specific needs and capabilities): - when people have psychological obstacles to recovery, - when previous evidence-based management has not been effective **No clear recommandation**: back school**Not recommended**: - belts and corsets - foot orthotics, rocker sole shoes - manual traction - ultrasounds - percutaneous electrical nerve stimulation - transcutaneous electrical nerve stimulation - interferential therapy **No clear recommandation**: acupuncture TOP [13]2015Canada**Recommended**: - acetaminophen - NSAIDs - muscle relaxants - tricyclic antidepressants - herbal medicines **Not recommended**: - selective Serotonin reuptake inhibitors - antibiotics (based on MRI Modic Changes) **No clear recommendation**: - opioids and tapentadol - marijuana (dried cannabis) - Duloxetine**Recommended**: capsaicin frutescens **No clear recommendation**: topical NSAIDs, Buprenorphine transdermal system**Not recommended**: prolotherapy as a sole treatment**No clear recommendation**: - prolotherapy as an adjunct - epidural steroid injections - therapeutic sacroiliac joint injections insufficient evidence - trigger point injections**Recommended**:exercise and therapeutic exercise:- initiate gentle exercise and gradually increase the exercise level within pain tolerance- may include unsupervised walking and group exercise programs - when exercise exacerbates pain, programme should be assessed by a qualified physical therapist - if exercise exacerbates pain, patients should be assessed by a physician - therapeutic aquatic exercise -Viniyoga and Iyengar types of yoga**Recommended**: massage therapy (as an adjunct to an active rehabilitation program)**Recommended**: provide brief education to optimize function- review of clinical examination results- provision of low back pain information and advice to stay active- reduce fear and catastrophizing**Recommended**:- when group chronic pain cognitive behavioral therapy programs are not available, consider referral for individual cognitive behavioral therapy - respondent behavioral therapies (progressive relaxation or EMG biofeedback)**Recommended**: - structured community-based self-management group program:- for patients interested in learning pain coping skills- most community-based programs also include exercise and activity programming- if not available: individual self-management counselling (trained professional) - multidisciplinary treatment program: after no improvement with primary care management**Recommended**: - acupuncture: short-term therapy or as an adjunct to a broader active rehabilitation program**Not recommended**: - motorized traction- transcutaneous electrical nerve stimulation (as a sole treatment)**No clear recommendation**: - manual therapy (spinal manipulative treatment or spinal mobilization)- therapeutic ultrasound - gravity tables (inverted traction, self-traction, gravitational traction) - ow-level laser therapy - mindfulness-based meditation - shock-wave treatment - spa therapy - back belts, corsets, - non-motorized traction- craniosacral massage/therapy - intramuscular stimulation - interferential current therapy - touch therapiesChenot [14]2017Germany**Recommended**: - NSAID - Metamizole **Not recommended**: - acetaminophen - Flupirtine - intravenously, intramuscularly or subcutaneously administered analgesic drugs, local anesthetics, glucocorticoids, or mixed infusions**No clear recommendation**: - COX-2-inhibitors: can be used if NSAIDs are contraindicated or poorly tolerated - opioids: ∘can be a treatment option for acute non-specific low back pain if non-opioid analgesics are contraindicated or have been ineffective∘regularly reassessed at intervals <4 weeks ∘to treat chronic non-specific low back pain for 4 to 12 weeks initially∘if this brief period of treatment brings an improvement in the pain while causing only minor or no side effects, opioid drugs can be a long-term therapeutic option

**Recommended**: - instruction to continue usual physical activities - rehabilitative sports and functional training - progressive muscle relaxation**No clear recommendation**: massage**Recommended**: - explain the condition and the treatment to the patient - encourage the pursuit of a healthful lifestyle, including regular physical exercise - patients should be advised against bed rest- initiation and coordination of psychotherapeutic care, if necessary- possibly social counseling**Recommended**: initiation and coordination of psychotherapeutic care, if necessary**Recommended**: - exercise therapy combined with educative measures based on behavioral-therapeutic principles should be used in the primary treatment of chronic non-specific low back pain - multimodal programs if less intensive evidence-based treatments have yielded an insufficient benefit: ∘multidisciplinary assessment∘stepwise reintroduction to the workplace or initiation of occupational reintegration measures **No clear recommendation**: - self-administered heat therapy - manual therapies (manipulation and mobilization)- ergotherapy - back school- acupuncture => could be used to treat chronic low back pain in combination with activating therapeutic measures**Not recommended**: - interference-current therapy - kinesiotaping - short-wave diathermy - laser therapy - magnetic field therapy - medical aids - percutaneous electrical nerve stimulation (PENS) - traction devices - cryotherapy - transcutaneous electrical nerve stimulation (TENS), - therapeutic ultrasoundHAS [2]2019France**First line**: acetaminophen, non-steroidal anti-inflammatory drugs (low dose, short duration); **Second line**: opioids (risk of misuse). Antidepressants and anticonvulsants are not indicated in acute LBP, possible use in case of chronic pain. **No opinion for** nefopam, cortico-steroids. **Not recommended**: muscle relaxants. **No indication for**: vitamin D, antibiotic, anti-TNF alpha.**No indication** for lidocaine patchGenerally no indication for LBP infiltration without root painPhysical exercise is the main treatment: self-management in first line: return to daily activities (and professional activities if possible), adapted physical activities and sports (progressive and fractional)**Suggested/Recommended:** physiotherapy (active participation of patient); patient education; mobilizations, manual therapy (only as part of a multimodal combination of treatments with supervised exercises and on second-line treatment)**Suggested/Recommended:** deliver reassuring information**Suggested/Recommended:** second-line treatment: cognitive behavioral therapy (only as part of a multimodal combination of treatments with supervised exercises)**Suggested/Recommended:** third-line treatment for patients with persistant pain and psychosocial risk factors or in case of failure of first- and second-line treatments**Not recommended**: ultra sound therapy; lumbar tractions; plantar orthosis **No clear recommendation**: acupuncture, acupressure, dry needling; sophrology; relaxation; mindfulness; hypnosis; lumbar brace; lumbar beltNICE [12]2016UK**Recommended**: - oral NSAIDs: - weak opioids (±acetaminophen): only if NSAID is contraindicated, not tolerated or has been ineffective. **Not recommended**:- acetaminophen alone - opioids - selective serotonin reuptake inhibitors, serotonin–norepinephrine reuptake inhibitors or tricyclic antidepressants - gabapentinoids or anticonvulsants
**Not recommended**: spinal injections in LBP **Recommended**: - radiofrequency denervation, to consider in chronic LBP: ∘when non-surgical treatment has not worked∘if the main source of pain is thought to come from structures supplied by the medial branch nerve∘for moderate or severe levels of localized back pain (5 or more on a visual analogue scale) ∘only in people with chronic low back pain after a positive response to a diagnostic medial branch block **Recommended**: group exercise program:- biomechanical, aerobic, mind–body or a combination of approaches - take people’s specific needs, preferences and capabilities into account when choosing the type of exercise
**Recommended**:- advice and information, tailored to their needs and capabilities, - help them self-manage low back pain - information on the nature of low back pain - encouragement to continue with normal activities**Recommended**:- psychological therapies using a cognitive behavioral approach - as part of a treatment package including exercise**Recommended**:- combined physical and psychological program∘when they have significant psychosocial obstacles to recovery ∘when previous treatments have not been effective - promote and facilitate return to work or normal activities of daily living**Recommended**:- consider manual therapy (spinal manipulation, mobilization or soft tissue techniques) as part of a treatment package including exercise**Not recommended**: - belts or corsets- foot orthotics- rocker sole shoes- traction- acupuncture- ultrasound- percutaneous electrical nerve simulation (PENS)- transcutaneous electrical nerve simulation (TENS)- interferential therapyQaseem [17]2017USA**Recommended** (in patients who have had an inadequate response to nonpharmacologic therapy): - first line: NSAIDs - second line: tramadol or Duloxetine - If failure: opioids if the potential benefits outweigh the risks

**Recommended:** exercise, Tai Chi, yoga**Recommended:**- motor control exercise- progressive relaxation

**Recommended**: multidisciplinary rehabilitation**Recommended**: - acupuncture- mindfulness-based stress reduction - electromyography biofeedback- low-level laser therapy, operant therapy - spinal manipulation (low-quality evidence)NASS [16]2020North America**Suggested/Recommended:** opioid pain medications (short duration)**Not recommended:** oral or IV steroids; antidepressants **No clear recommendation**: anticonvulsants; vitamin D; selective NSAIDs**Suggested/Recommended:** topical capsaicin **No clear recommendation**: lidocaine patch**No clear recommendation**: caudal epidural steroid injections; interlaminar epidural steroid injections; zygapophyseal joint injection; intradiscal steroids; intradiscal platelet rich plasma**Suggested/Recommended:** yoga; aerobic exercise**Suggested/Recommended:** McKenzie method**No recommended**: traction; ultrasound; addition of massage to an exercise program; lumbar stabilization **No clear recommendation**: transcutaneous electrical nerve stimulation (TENS); dry needling**Suggested/Recommended:** back school**No clear recommendation**: patient education**Suggested/Recommended**: cognitive behavioral therapy (in combination with physical therapy)treatments targeting fear avoidance (combined with physical therapy)

ACOEM [23]2020USA**Recommended:**- NSAIDs- acetaminophen - antidepressants- skeletal muscle relaxants: for acute exacerbations of chronic LBP **Not recommended:** opioids, antibiotics, antidepressants,anticonvulsants,bisphosphonates, calcitonin and oral and intravenous colchicine NMDA receptor/antagonists skeletal muscle relaxants glucocorticosteroidsTNF-a vitamin supplementation **No Clear recommendation**: Thiocolchicoside**Recommended:** capsaicin**Not recommended:**
- lidocaine patches - Spiroflor - DMSO, N-acetylcysteine, EMLA, and wheatgrass cream **No Clear recommendation**: topical NSAIDs or other creams 
**Recommended:**- exercise prescription- self-administered or enacted through formal therapy appointments- aerobic exercises (progressive walking program) - directional exercises which centralize or abolish the pain - slump stretching exercises 3 to 5 times a day- strengthening exercises - specific strengthening exercises - yoga and tai chi for select, motivated patients**Not recommended:**- stretching exercises in the absence of significant range of motion deficits - abdominal strengthening exercises as a sole or central goal of a strengthening program **No Clear recommendation:** Pilates**Recommended:**- massages for select use as an adjunct to more efficacious treatments (aerobic and strengthening exercise program)- self-applications of low-tech heat therapies and cryotherapies- aquatic therapy for select chronic LBP patients (extreme obesity, significant degenerative joint disease, etc.)**Not recommended**: - mechanical devices for administering massage - reflexology - high-tech devices of heat and/or cryotherapy - diathermy- lumbar extension machines**No Clear recommendation:** myofascial release**Recommended:**- maintaining maximal levels of activity, including work activities, - work modifications should be tailored taking into consideration 3 main factors: (1) job physical requirements; (2) severity of the problem; and (3) the patient’s understanding of his or her condition- fear avoidance belief training for patients with elevated fear avoidance beliefs**Not recommended:** bed rest

**Recommended:**- lordotic sitting posture - sleep posture comfortable- manipulation or mobilization (component of an active exercise program)- acupuncture - transcutaneous electrical nerve simulation TENS **Not recommended:**- specific beds or other commercial sleep products- kinesiotaping - shoe lifts or insoles except for individuals with significant leg length discrepancy of more than 2 cm - lumbar supports - magnets - traction - low-level laser therapy- microcurrent electrical stimulation**No Clear recommendation**: - specific mattresses, bedding, and water bed- medical foods (Ther- amine, an amino acid formulation) - herbal- iontophoresis- inversion therapy- infrared therapy- ultrasoundsVA/DOD [4]2017USA**Recommended**: - NAIDs- duloxetine- non-benzodiazepine muscle relaxant for acute exacerbations of chronic low back pain **Not recommended**: - non-benzodiazepine muscle relaxant- benzodiazepines - oral or intramuscular injection corticosteroids- long-term opioid therapy - chronic use of oral acetaminophen **No clear recommendation**: time-limited opioid therapy, for acute exacerbations of chronic low back paintime-limited (less than 7 days) acetaminophen therapyanticonvulsants nutritional, herbal, and homeopathic supplements**No clear recommendation**: topical preparations**Not recommended**: - spinal epidural steroid injections - intra-articular facet joint steroid injections**Recommended**: - clinician-directed exercises - exercise program, which may include Pilates, yoga, and tai chi
**Recommended**: - provide evidence-based information with regard to their expected course - advise patients to remain active, - provide information about self-care options- add structured education component as part of multicomponent self-management intervention**Recommended**: - cognitive behaviral therapy **Recommended**: - Multidisciplinary or interdisciplinary rehabilitation programme which should include at least one physical component and at least one other component of the biopsychosocial model (psychological, social, occupational)- for selected patients not satisfactorily responding to more limited approaches**Recommended**: - spinal mobilization/manipulation as part of a multimodal programme - acupuncture - mindfulness-based stress reduction.**No clear recommendation**: - lumbar supports- ultrasound - transcutaneous electrical nerve stimulation (TENS)- lumbar traction- electrical muscle stimulation- medial branch blocks

*Physiotherapy.* Physiotherapy represents a first-line treatment for chronic LBP or patients with risk factors for chronic LBP [2]. Rehabilitation techniques are not always detailed. Massages and mobilization of soft tissues are recommended in most guidelines, but only as part of multimodal treatment with active rehabilitation. The addition of massage to an exercise program provides no benefit when compared to an exercise program alone [16]. Other techniques are mentioned (i.e., transcutaneous electrical nerve stimulation, manual therapy, McKenzie method) but there is no consensus. Tractions are not recommended [16].

*Psychological treatment.* Cognitive behavioral therapy is recommended, in combination with physical therapy, to improve pain levels in patients with LBP, and to improve functional outcomes and return to work [2,12,13,16,18]. Treatments that target fear avoidance, combined with physical therapy to improve LBP in the first 6 months, may also be offered [16], as well as mindfulness-based stress reduction approaches [4].

*Patient education.* On the issue of educating patients with chronic LBP, all guidelines agree on the maintenance of maximal levels of activity, and promoting and facilitating a return to work or normal daily living activities as soon as possible [2,12,13,14,15,18]. Some guidelines recall the importance of informing patients on the nature of LBP based on data from evidence-based medicine [4,12,14]. The 2021 French guidelines insist on the importance of reassuring patients [2], and the ACOEM underlines the importance of interventions targeting erroneous fears and beliefs. Some guidelines recommend providing advice and information to enhance self-management [4,12,13,18].

*Multidisciplinary rehabilitation.* The international guidelines agree on the interest of multidisciplinary rehabilitation to manage LBP patients when they have psychological obstacles to recovery after there is no improvement with primary care management. Multidisciplinary rehabilitation programs should include at least one physical component, and at least one other component of the biopsychosocial model (psychological, social and occupational). The content of the programs is not always detailed, and varies from one country to another. The physical component of the multidisciplinary rehabilitation program is based on exercise and activity to promote and facilitate a return to work or normal activities of daily living. There is no consensus on the psychological component. Various approaches are suggested: cognitive behavioral approaches [18], learning pain coping skills [13] and mindfulness-based stress reduction [17]. The German guidelines underline the usefulness of a patient’s multidisciplinary assessment [14].

*Other non-pharmacological treatments.* The international guidelines suggest that other therapeutic options corresponding to adjuvant therapy may be carried out. Acupuncture and manual therapy are the two treatments that almost all of the guidelines agree on. Acupuncture-based therapy in the management of patients with LBP is often reported as a short-term therapy, or as an adjunct to a broader active rehabilitation program [2,4,13,15,17], and is suggested to be cost-effective when compared with other medical/interventional treatments [16]. In the same way, manual therapy (spinal manipulation, mobilization or soft tissue techniques) is considered as part of a treatment package that includes exercise [2,4,12,15,17]. However, for patients with chronic LBP, there is conflicting evidence that outcomes for spinal manipulative therapy are clinically different than no treatment, medication or other modalities [16]. A long list of other treatments is mentioned, but does not lead to specific recommendations for clinical practice.

#### 3.3.2. Pharmacological Treatments

The pharmacological treatments indicated for pain relief, especially during acute flare-up, are numerous.

*Oral treatments.* There is a consensus on the use of oral non-steroidal anti-inflammatory drugs (NSAIDs), which are almost systematically recommended as a first-line treatment, taking into account the toxicity and the person’s risk factors, and respecting the rule of “the lowest effective dose for the shortest possible period”. Only the NASS reports that there is insufficient evidence to make a recommendation for or against the use of selective NSAIDs for the treatment of LBP [16]. Weak opioids are recommended as a second-line treatment, in association with acetaminophen or not, when NSAIDs are contraindicated, not tolerated or have failed. In contrast, there is no consensus concerning the use of acetaminophen. There are also controversies regarding the use of antidepressants, opioids, anticonvulsants and muscle relaxants.

*Topical treatments.* There is no consensus for the use of the lidocaine patch in the treatment of LBP. The NASS iterates that there is insufficient evidence to make a recommendation for or against the use of the lidocaine patch in this indication [16]; the ACOEM and the 2021 French guidelines are in favor of not using the lidocaine patch [2,15]; and the Canadian guidelines recommend its use [13]. Topical capsicum is recommended as an effective treatment for LBP over a short period [15,16]. Concerning topical NSAIDs, the recommendations are not clear [13,15,18]; however, most of the recommendations do not endorse any topical treatment in questioning the relevance of a topical treatment in he chronic LBP [4,12,14,17].

*Spinal injections.* Epidural steroid injections are generally not recommended for patients who are not suffering from root pain. Facet joint infiltration is not recommended in chronic LBP. However, the NASS suggests that intra-articular steroid joint injections may be considered in patients with suspected sacroiliac joint pain, and intradiscal steroid injections are suggested to provide short-term improvements in pain and function in patients with Modic changes, but concludes there is insufficient evidence that intradiscal steroids improve pain or function in patients with discogenic LBP.

## 4. Discussion

Most LBP patients experience self-limited episodes of pain, with improvements occurring within the first month. However, in 6 to 8% of patients, LBP can become chronic [ref]. New concepts in the 2021 French guidelines of “recurrence of LBP” and “LBP at risk for chronicity” (yellow flags) highlight that contributors to both causes and consequences of LBP include pathoanatomical factors, but also contextual and psychosocial factors. Even in clinical practice, with composite questionnaires and searches for yellow, blue and black flags to evaluate fears and beliefs, psychological and social contexts are probably tedious, and it seems essential to evaluate with simple questions the social, professional and thymic repercussions of LBP. Interestingly, fear-avoidance beliefs not only affect patients [24], but also physicians [25]. Indeed, fears and beliefs of general practitioners can also negatively influence their ability to follow guidelines concerning physical and occupational activities for patients with chronic LBP, despite educational sessions on LBP [25].

The diagnosis of “non-specific LBP” assumes that in the absence of a readily identifiable plausible nociceptive source or known pathoanatomical cause, there is none [3,12,17], and that carrying out clinical or imaging investigations is of little value, and may even cause harm. In the 2021 French guidelines, the new definition of LBP, defined as “degenerative/non degenerative/unrelated to anatomical lesions” [2], reflects that the advances in pathoanatomical understanding are vital to address the causes of non-specific LBP, in order to better understand origins of the symptoms, offer targeted and effective treatments, evaluate prognoses and prevent chronicization [5,6]. This new classification of chronic LBP has been taken up by spine surgeons who recognize that patients classified as “non-specific LBP” constitute an extremely heterogeneous population, in whom neither the causal anatomical lesions nor the abnormalities in spinal alignment were taken into account [26].

The diagnostic approach is well codified in acute LBP, but not in chronic LBP; the reason for this is probably because of the lack of valid “diagnostic biomarkers” in the absence of a reliable gold standard. Indeed, there are no published data that have found a specific history or physical examination that would indicate structures that cause the pain [27]. Interestingly, the Chinese Association for the Study of Pain reported simple questions about pain (duration, location, factors that worsen or improve pain, etc.) and physical examination (spine deformity, local condition, tenderness, percussion pain, Lasègue sign, etc.) to phenotype non-specific LBP into discogenic LBP, zygapophyseal joint pain, sacroiliac joint pain and soft tissue-derived LBP [28].

As a result of frequent anatomo-clinical discrepancies, the international guidelines agree that there is insufficient evidence to make a recommendation for or against obtaining imaging in the absence of red flags in chronic LBP. Indeed, studies in the asymptomatic population report a significant number of abnormalities [29]. Only the 2021 French guidelines suggest using MRIs in chronic LBP lasting more than 3 months [2]. In the ACOEM practice guidelines, it was clearly concluded that diagnostic testing is not indicated for the majority of people with LBP [23]. Furthermore, even when a readily identifiable plausible nociceptive source is present, people with chronic LBP may have more than one cause of LBP; hence, phenotyping should also include dimensions of functioning other than pathoanatomy [5,6]. Altogether, these findings reflect the limitations of how popular diagnostic investigations, history, clinical tests and imaging are used, all of which lack specificity when considered in isolation [6].

There are published data that reported the efficacy of glucocorticoid intradiscal injections for people with chronic LBP and active discopathy (Modic 1 changes) [30,31,32], confirming the significance of Modic 1 changes as an imaging biomarker of a painful intervertebral disc when considered with clinical and biological biomarkers [33,34]. This may serve as a model of validation for phenotypes in people with LBP, and the phenotyping assistance provided by MRI [5].

Concerning therapeutical approaches, a study of international recommendations led to the observation of a common philosophy, without real homogeneity in the practice guidelines.

Indeed, if the use of oral NSAIDs and the practice of exercise therapy and physical activity reach consensus, just as cognitive behavioral therapy in combination with physical therapy for chronic LBP patients with risk factors of chronicity (multidisciplinary rehabilitation programs), other treatments are still under debate. This is the case for antidepressants, opioids, anticonvulsants and muscle relaxants. Moreover, the content of multidisciplinary care is not always clearly specified. Thus, each country seems to compose its care programs more from habits of practice than from scientific evidence.

Our review has limitations. Our narrative review does not allow for drawing conclusions about the hierarchy of treatments. A network meta-analysis would be more appropriate to address this specific point. The cause of cLBP may change the treatment (pharmacological or not pharmacological). However, in most selected guidelines, individuals were selected under the umbrella of “now-specific” cLBP, which lacks sufficient granularity to address this point. Some treatments, such as acupuncture, were not reviewed.

## 5. Conclusions

Investigating the causes of chronic LBP is a challenge in daily practice and in research, because the pathogenesis of chronic LBP includes pathoanatomical factors as well as psychosocial factors. History and clinical and imaging testing lack specificity when considered in isolation; straightforward methods to fully validate their value as diagnosis and/or prognosis biomarkers are lacking [5]. However, recent advances in clinical semiology, imaging techniques, and the elucidation of spinal biomechanics, have shed new light on chronic LBP; these data are probably not highlighted enough in the international recommendations. For several decades, chronic LBP has remained a public health problem, and is the leading cause of disability worldwide in young adults. Despite the publication of numerous scientific recommendations on the subject, no improvement in the situation has been observed. The authors question the place of the more precise etiological diagnosis called phenotyping. Indeed, this review finds a lack of precision in the phenotyping of patients with chronic LBP. Nevertheless, the emergence of patient phenotyping in certain publications should be noted. It now appears necessary to better phenotype LBP patients, in order to be able to offer more targeted therapies and improve the effectiveness of treatments.

## Figures and Tables

**Figure 1 jcm-12-01685-f001:**
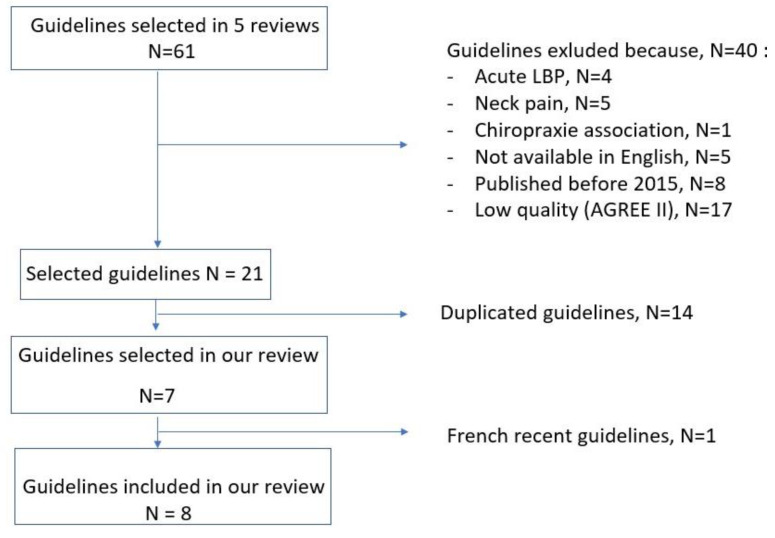
Flow diagram.

## Data Availability

The full original protocol and dataset can be accessed upon request for academic researchers by contacting Christelle Nguyen (christelle.nguyen2@aphp.fr).

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
