# Peer review of "Chronic Low Back Pain: A Narrative Review of Recent International Guidelines for Diagnosis and Conservative Treatment"

_jcm, 2023, doi:10.3390/jcm12041685_

Round 1
Reviewer 1 Report
Reviewer Comments
Thank you very much for the opportunity to review the manuscript submission entitled: Chronic low back pain: a narrative review of recent international guidelines for the diagnosis and conservative treatment.
The current paper aims to provide a critical overview of current international recommendations regarding the management of non-specific CLBP. The review is interesting, and it has a relevant rationale. However, some limitations and constructive comments are pointed out below:
General comments:
- The paper requires a deep English language review. The flow of sentences and word choices are not according to Plain English Standards. Many sections are difficult to understand
Specific comments
Title and Abstract
· The title is good.
· The abstract should focus on implications for future research, and clinical practice
Introduction
· The scientific background and rationale for the narrative review need to be emphasized.
· Describe the rationale for the review in the context of what is already known.
· Specify the key questions identified for the review topic.
Methods
· Elaborate and specify the process for identifying the literature search (eg, publication status, study design, and databases of coverage).
Discussion
· The discussion should be emphasized the limitations, quality of research reviewed, and need for future research.
Reviewer 2 Report
In this narrative review the authors evaluated the guidelines on CLB pain.
However the number of the authors is very high considering their skill, only CD, EL, L-C MM, RF, AR and NC published some manuscripts on low back pain.
1) Methods: please clarify the numbers in methods, in fact selected guidelines=21, duplicated=14, selected in review=8, but probably it may be 7.
2) Please add a synthetic table were you show which could be the better treatment
3) please, in discussion indicate if the cause of low back pain could change the treatment (pharmacological or not pharmacological)
4) I not understand why you excluded other treatments, in fact to date several patients used acupuncture therefore I think that it could be an add on therapy
5) please indicate in discussion the role of physical activity
Round 2
Reviewer 1 Report
The authors have addressed all the comments. Can be accepted for publication in the current form.
Reviewer 2 Report
none